# Genetic Diversity of the Fall Armyworm *Spodoptera frugiperda* (J.E. Smith) in the Democratic Republic of the Congo

Matabaro Joseph Malekera [1,2], Damas Mamba Mamba [2], Gauthier Bope Bushabu [2], Justin Cishugi Murhula [2], Hwal-Su Hwang [1,3] and Kyeong-Yeoll Lee [1,3,4,*]

[1] Department of Plant Medicine, College of Agriculture and Life Sciences, Kyungpook National University, Daegu 41566, Republic of Korea; jmatabaro@live.com (M.J.M.); bgtwo2@naver.com (H.-S.H.)

[2] Department of Plants Protection, Ministry of Agriculture, Kinshasa 8722, Democratic Republic of the Congo; damasmamba@yahoo.fr (D.M.M.); gauthierbush2009@yahoo.fr (G.B.B.); jcishugim@gmail.com (J.C.M.)

[3] Research Institute for Dok-do and Ulleung-do Island, Kyungpook National University, Daegu 41566, Republic of Korea

[4] Institute of Plant Medicine, Kyungpook National University, Daegu 41566, Republic of Korea

* Correspondence: leeky@knu.ac.kr; Tel.: +82-53-950-5759

**Abstract:** In 2016, the fall armyworm (FAW), *Spodoptera frugiperda*, invaded western Africa and rapidly spread in sub-Saharan Africa, causing significant losses in yields of corn, a major food crop in Africa. Although the Democratic Republic of the Congo (DRC) is a large corn-growing country, the impact of FAW has not been investigated. This study was designed to expand investigations on the genetic diversity of FAW populations in the DRC. We collected FAW individuals from eight provinces across the country, for analysis of genetic variation. Based on the partial sequences of both mitochondrial *cytochrome oxidase subunit I* (*COI*) and nuclear *triosephosphate isomerase* (*Tpi*) genes, we compared polymorphic features of the *COI* haplotype and *Tpi* single nucleotide polymorphisms. The results revealed that most (84%) of the analyzed individuals were heterogeneous hybrids *Tpi*-corn/*COI*-rice (*Tpi*-C/*COI*-R), whereas 16% were homogenous *Tpi*-corn/*COI*-corn (*Tpi*-C/*COI*-C). Further analysis of the fourth exon/intron sequences of the *Tpi* gene identified two subgroups, *Tpi*Ca1 and *Tpi*Ca2, constituting 80% and 20%, respectively, of the collected individuals. Analysis of genetic variation among native and invasive populations indicated significant genetic differences (10.94%) between the native American and DRC populations, whereas both the DRC and African populations were genetically closer to Asian than American populations. This study provides important information on FAW genetic diversity in the DRC, which can be used for effective management of FAW.

**Keywords:** climate change; fall armyworm; genetic diversity; invasive pest; quarantine





## 1. Introduction

The fall armyworm (FAW), *Spodoptera frugiperda* (Lepidoptera, Noctuidae), is a devastating agricultural pest in the tropical and subtropical regions [1,2]. Although FAW is native to south America, since the first outbreak outside its native region in 2016, its global distribution range has swiftly expanded throughout Africa, Asia, and recently, Oceania [3–5]. This rapid range expansion is probably due to its capacity to adapt to a wide range of temperature conditions and its polyphagy [6,7]. FAW is a highly polyphagous species that can feed on at least 76 plant families, mainly Poaceae, Asteraceae, and Fabaceae [8]. However, FAW has a strong propensity for corn (Poaceae), which is the main food crop for more than 200 million African smallholder farmers [2]. FAW larvae damage corn plants by feeding on leaves, stems, and reproductive parts, thus destroying their growth potential [9]. When their population is large, they develop an "armyworm" behavior and disperse in large numbers, attacking almost all vegetation in their path [10].

According to host plant preference, FAW consists of two strains, namely, the corn strain and rice strain [11]. These strains are morphologically similar but genetically different with

2.09% genome divergence. They also exhibit variation in developmental, physiological, and ecological features such as host plant preference, and sex pheromones [11–13]. The rice strain is typically associated with rice *Oryza sativa* L., sugar cane *Saccharum officinarum* L., and grass species, such as Johnson grass *Sorghum halepense* (L.), Bermuda grass *Cynodon dactylon* (L.), whereas the corn strain is associated with corn *Zea mays* L. sorghum *Sorghum bicolor* (L.), soybean (*Glycine max*), and cotton *Gossypium hirsutum* (L.) [13,14].

Considering the intra- and interspecific variation among FAW strains, reliable strain identification is essential in field studies of FAW populations. The two strains of FAW are identified mainly based on polymorphisms in the mitochondrial gene *cytochrome c oxidase subunit 1* (*COI*) and the nuclear gene *triosephosphate isomerase* (*Tpi*) [15,16]. In the western hemisphere, the relationship between the *COI* and *Tpi* markers is important for identifying FAW strains [17]. However, in the invaded regions of Africa and Asia, strain identification using the two markers has shown discordant results. Overall, the host association in invasive populations was accurately predicted by *Tpi* but not *COI* [18,19]. The discordance between the *COI* and *Tpi* markers indicates the hybrid nature of FAW populations that invaded Africa [17]. This hybrid nature of invasive populations was lately confirmed by whole-genome sequencing studies [20,21]. The identification of two *COI*-based haplotypes and a small number of *Tpi* haplotypes showed that genetic diversity was low in the invasive populations of FAW [17,22,23]. The low genetic diversity and small number of haplotypes observed in most invaded locations indicate a possible recent introduction from a common source of the FAW population and could affect the monitoring of the crops at risk in these locations [17,19,23].

This study was conducted to expand investigations on the genetic diversity of FAW populations in the DRC. Thus, additional samples collected in new locations in the DRC were combined and compared with those from earlier studies [19,22] to provide a more representative picture of the country-wide genetic structure of FAW in the DRC. The genetic characterization of FAW in the DRC during the first six years of invasion can predict changes in the populations as they rebalance and respond to pest management measures. Additionally, the comparison of the populations of FAW in the DRC with those from both native and other invaded regions can provide the phylogeographic patterns and relationships of FAW haplotypes in the DRC and could be used in understanding the possible route of the FAW population that invaded the DRC.

## 2. Materials and Methods

### 2.1. Collection of FAW Samples

Samples were collected from corn fields at 21 locations in 8 provinces in the DRC during the five-year period from 2018 to 2022 (Table 1). After field collection, the larvae were preserved individually in 1.5 mL microcentrifuge tubes in 70% ethanol. The samples were transported to the plant clinic in Kinshasa, DRC and kept at −20 °C until they were sent to the Laboratory of Insect Molecular Physiology at Kyungpook National University, Republic of Korea, for DNA extraction and further genomic analysis.

### 2.2. DNA Extraction

Total genomic DNA was extracted from the head of single larva using a pure link genomic DNA mini kit (Invitrogen, Carlsbad, CA, USA) [22]. Each sample was homogenized in a 1.5 mL microcentrifuge tube containing 200 μL of digestion buffer and 20 μL of proteinase K (50 μg/mL) before it was incubated at 56 °C for 30 min. The DNA supernatant was collected in a genomic spin column and stored in a new 1.5 mL microcentrifuge tube at −20 °C until downstream analysis. DNA quality and concentration were assessed using a NanoPhotometer™ (Implen GmbH, Schatzbogen, Germany). The remaining portions of the samples were kept in 70% ethanol at −20 °C.

**Table 1.** List of fall armyworm *Spodoptera frugiperda* samples and the locations from which they were collected from the DRC.

| No. | Sample ID | Province/Territory/Village | Location | Collection Date (Day/Month/Year) | Accession Number | | Genetic Group | |
|---|---|---|---|---|---|---|---|---|
| | | | | | *COI* | *Tpi* | *COI* | *Tpi* |
| 1 | Congo11 | Sud-Kivu/Kabare/ Katana | 2°22′51″ N 28°82′35″ E | 29 November 2018 | MT103350 | MT894220 | *COI*-RS | *Tpi*-Ca1a |
| 2 | Congo42 | Sud-Kivu/Walungu/Nduba | 2°63′73″ N 28°69′63″ E | 15 December 2018 | MT103349 | MT894225 | *COI*-RS | *Tpi*-Ca1a |
| 3 | Congo3 | Sud-Kivu/Kalehe/Bunyakiri | 1°99′49″ N 28°54′62″ E | 29 November 2018 | OQ612484 | OQ632453 | *COI*-RS | *Tpi*-Ca1a |
| 4 | Congo41 | Sud-Kivu/Uvira/Sange | 3°06′10″ N 29°08′55″ E | 15 December 2018 | MT933055 | MT894224 | *COI*-RS | *Tpi*-Ca2b |
| 5 | Congo31 | Sud-Kivu/Uvira/Luvungi | 2°89′15″ N 28°97′12″ E | 15 December 2018 | MT933054 | MT894223 | *COI*-RS | *Tpi*-Ca2a |
| 6 | Congo21 | Sud-Kivu/Kalehe/Minova | 1°74′73″ N 28°98′78″ E | 29 November 2018 | MT933053 | MT894222 | *COI*-RS | *Tpi*-Ca2a |
| 7 | Congo12 | Sud-Kivu/ Kabare/Miti | 2°33′06″ N 28°76′69″ E | 29 November 2018 | MT933052 | MT894221 | *COI*-RS | *Tpi*-Ca2b |
| 8 | K1 | Lomami/Kabinda/Kabinda | 6°07′48″ S 24°28′48″ E | 18 July 2020 | OP132901 | OQ468459 | *COI*-RS | *Tpi*-Ca1a |
| 9 | Gem1 | Sud-ubangi/Gemena/Gemena1 | 3°14′56″ N 19°46′36″ E | 15 July 2020 | OP132892 | OQ468451 | *COI*-RS | *Tpi*-Ca1a |
| 10 | Bkd | Sud-ubangi/Gemena/Bokunda | 3°12′39″N 19°46′29″ E | 15 July 2020 | OP132899 | OQ468460 | *COI*-RS | *Tpi*-Ca1a |
| 11 | Bsg1 | Sud-ubangi/Gemena/Bosengwen | 3°13′50″N 19°42′57″ E | 18 July 2020 | OP132898 | OQ468458 | *COI*-CS | *Tpi*-Ca1a |
| 12 | Bbw1 | Sud-ubangi/Gemena/Bombawuli | 3°13′48″ N 19°53′51″ E | 18 July 2020 | OP132896 | OQ468455 | *COI*-RS | *Tpi*-Ca1a |
| 13 | Mtf1 | Tanganyika/Kalemie/Kalemie | 5°52′08″ S 29°10′14″ E | 21 July 2020 | OP132894 | OQ468453 | *COI*-RS | *Tpi*-Ca1a |
| 14 | Tshb1 | Tshuapa/Boende/Boende1 | 0°17′13″ S 20°52′24″ E | 18 July 2020 | OP132895 | OQ468454 | *COI*-RS | *Tpi*-Ca1a |
| 15 | Blk1 | Tshuapa/Boende/Baliko | 0°18′05″ S 20°52′30″ E | 18 July 2020 | OP132897 | OQ468456 | *COI*-CS | *Tpi*-Ca1a |
| 16 | Bde1 | Tshuapa/Boende/Boende3 | 0°16′39″ S 20°53′05″ E | 15 July 2020 | OP132898 | OQ468457 | *COI*-RS | *Tpi*-Ca1a |
| 17 | Isi1 | Haut-Uélé/Isiro/Isiro | 2°45′57″ N 27°36′32″ E | 8 August 2020 | OP132893 | OQ468452 | *COI*-RS | *Tpi*-Ca1a |
| 18 | M1 | Kongo central/Matadi/Matadi | 5°47′58″ S 13°26′26″ E | 18 July 2020 | OP132900 | OQ632454 | *COI*-RS | *Tpi*-Ca1a |
| 19 | Kst1 | Kongo central/Kisantu/Kisantu1 | 5°13′82″ S, 15°09′08″ E | 15 December 2022 | OQ427278 | OQ468462 | *COI*-RS | *Tpi*-Ca2a |
| 20 | Kst2 | Kongo central/Kisantu/Kisantu2 | 5°13′82″ S, 15°09′08″ E | 15 December 2022 | OQ427279 | OQ468466 | *COI*-CS | *Tpi*-Ca1a |
| 21 | Kst3 | Kongo central/Kisantu/Kisantu3 | 5°13′82″ S, 15°09′08″ E | 15 December 2022 | OQ427280 | OQ857569 | COI-RS | Tpi-Ca1a |
| 22 | Plaba1 | Kinshasa/Plateau de Bateke1 | 4°20′72″ S, 15°84′48″ E | 20 December 2022 | OQ427282 | OQ468463 | *COI*-RS | *Tpi*-Ca1a |
| 23 | Plaba2 | Kinshasa/Plateau de Bateke2 | 4°20′72″ S, 15°84′48″ E | 20 December 2022 | OQ427284 | OQ468464 | *COI*-CS | *Tpi*-Ca1a |
| 24 | Kimw1 | Kinshasa/Kimwenza1 | 4°47′11″ S, 15°30′14″ E | 20 December 2022 | OQ427281 | OQ468461 | *COI*-RS | *Tpi*-Ca1a |
| 25 | Kimw2 | Kinshasa/Kimwenza2 | 4°47′11″ S, 15°30′14″ E | 20 December 2022 | OQ427283 | OQ468465 | *COI*-RS | *Tpi*-Ca1a |

### 2.3. PCR Amplification and Sequence Analysis

DNA was subjected to PCR amplification using a SimpliAmp 96-Well Thermal Cycler (Applied Biosystems, Foster City, CA, USA). Each PCR reaction mixture of 30 μL contained 15 μL of Solg 2 × Taq PreMix (Solgent, Daejeon, Republic of Korea), 2 μL of each primer (10 pmol/μL), 2 μL of the DNA template, and 9 μL of sterile water. Partial *COI* (658 bp) and Tpi (444 bp) barcode regions of the FAW were amplified using the primer pairs LCO1490 (5′-GGTCAACAAATCATAAAGATATTGG-3′) and HCO2198 (5′-TAAACTTCAGGG TGACCAAAAAATCA-3′) for COI, and TPI412F (5′-CCGGACTGAAGG TTATCGCTTG-3′) and TPI1140R (5′-GCGGAAGCATTC GCTGACAACC-3′) for *Tpi* [23,24]. The thermo-cycling conditions for *COI* included an initial denaturation at 92 °C for 5 min, followed by 35 cycles of denaturation at 92 °C for 1 min, annealing at 55 °C for 1 min, and extension at 72 °C for 1 min. The *Tpi* gene thermo-cycling parameters included an initial denaturation at 94 °C for 1 min, followed by 33 cycles of denaturation at 92 °C for 30 s, annealing at 56 °C for 45 s, extension at 72 °C for 1 min, and final extension at 72 °C for 5 min. The amplified products were stained with ethidium bromide before they were visualized on 1% agarose gel under ultraviolet (UV) light. The amplified products were sequenced using the BigDye® Terminator Cycle Sequencing Kit and ABI Prism 3730XL DNA Analyzer (50 cm capillary) (DNA Sequencer) at the Celemics Sequencing Facility (Celemics, Seoul, Republic of Korea). The sequences generated in this study showed 100% similarity to those of FAW in the NCBI database. The sequences were submitted to the NCBI GenBank under accession numbers assigned to each specimen (Table 1).

### 2.4. DNA Polymorphism Analysis

*COI* sequences from our previous study [22] were aligned with the sequences generated in this study using ClustalW [24] and used for characterizing genetic diversity (Table 1). Furthermore, the diversity of the Congolese FAW population was compared with that from other geographic locations. The *COI* sequences reported from Africa (89), Asia (72), and America (126) (Table A1) were retrieved from GenBank database and trimmed to a length of 483 bp as this region was present in most the of sequences and was used for comparative polymorphism studies [25]. The dataset was classified into four main geographical categories: (1) Africa, (2) Asia, (3) America, and (4) the DRC. Descriptive statistics including nucleotide diversity, number of haplotypes (H), haplotype diversity (Hd), genetic neutrality, and mismatch distribution were generated using DnaSP ver. 6.12.03 [26]. Mismatch distribution curves which report the frequency of pairwise nucleotide-site differences between FAW sequences from the DRC, were generated using the constant population size model in DnaSP to further examine the demographic pattern of FAW in DRC.

The FAW *COI* and *Tpi* gene polymorphisms of the DRC samples were analyzed using previously published strain defining loci and polymorphic sites [27,28]. The single nucleotide polymorphisms (SNPs) (mCOI72, mCOI117, mCOI171, mCOI207, mCOI258, mCOI564, mCOI570, mCOI600, mCOI634, and mCOI663) that define the strain polymorphic sites of FAW found in the barcode region of the *COI* were used to distinguish between corn and rice strains of FAW in our previous study [22]. Additionally, the *Tpi* partial gene segment (444 bp), which contained 166 bp of the fourth exon (*Tpi*-E4) and 278 bp of the fourth intron (*Tpi*-I4), was used to identify the *S. frugiperda* host strain. The presence of the nucleotide base letters "C" or "T" at gTpi183, for the corn strain (*Tpi*-C) or rice strain (*Tpi*-R), respectively, allowed us to distinguish the FAW *Tpi*-based host strains.

### 2.5. Haplotype Network Plot and Phylogenetic Analysis

A haplotype network was constructed using the popART software ver. 1.7 [29]. Sequences were aligned and grouped within the four geographical regions using ClustalW [24] and Dnasp, respectively. The median joining network method was used to infer haplotype relationships. To generate the evolutionary relationship between the DRC FAW haplotypes, a phylogenetic tree for the *COI* gene was constructed using the maximum likelihood

method implemented in MEGA 6.0 [30], with other *Spodoptera* species and FAW corn and rice strains retrieved from NCBI [31,32]. For each phylogeny, 1000 bootstrap replicates were used to assess robustness using the Hasegawa–Kishino–Yano (HKY850) model and gamma distribution rate of variation between sites [33].

### 2.6. Analysis of Molecular Variance (AMOVA)

AMOVA was performed using Arlequin ver. 3.5.2.2 [34]. The analysis was conducted with four geographic groups including the rest of Africa, Asia, America, and the DR Congo. Apart from the overall AMOVA, the *COI* sequences from the four geographical regions were examined in six combinations comprising DRC vs. America, DRC vs. Africa, DRC vs. Asia, America vs. Africa, America vs. Asia, and Africa vs. Asia. We observed variance differences among groups by a randomized test with 1000 permutations in a haplotype-based standard AMOVA.

## 3. Results

### 3.1. PCR Amplification and Sequence Analysis

We recovered 25 nucleotide sequences of both *COI* and *Tpi* genes from the represented FAW samples collected from 21 different regions of the DRC (Table 1). Sequence analysis of the partial *COI* fragment (658 bp) showed that the *COI*-R constituted 84%, whereas the *COI*-C constituted 16% of the sequences (Figure 1b). Additionally, analysis of *Tpi* sequences on the polymorphic locus gTpi183 (which was used to identify the rice and corn strains) and at the *Tpi*-E4 was C but not T, which indicated that all the samples were from the *Tpi*-C genetic group. Furthermore, two *Tpi*-based haplotypes were identified in the DRC's FAW population, including the *Tpi*-Ca1 homozygous (in 80% of individuals), and *Tpi*-Ca2 homozygous (in 20% of individuals) (Figure 1c). The *Tpi*-R haplotype was not detected in any of the samples. Further analysis of the *Tpi*-Ca2 subgroup showed that the *Tpi*-Ca2a and *Tpi*-Ca2b genetic groups were detected in three and two individuals, respectively. Our results indicated that the nuclear *Tpi* marker consistently identifies the phenotypic feeding behavior of FAW on corn, which is the host plant of FAW in the DRC as well as in other African and Asian countries.

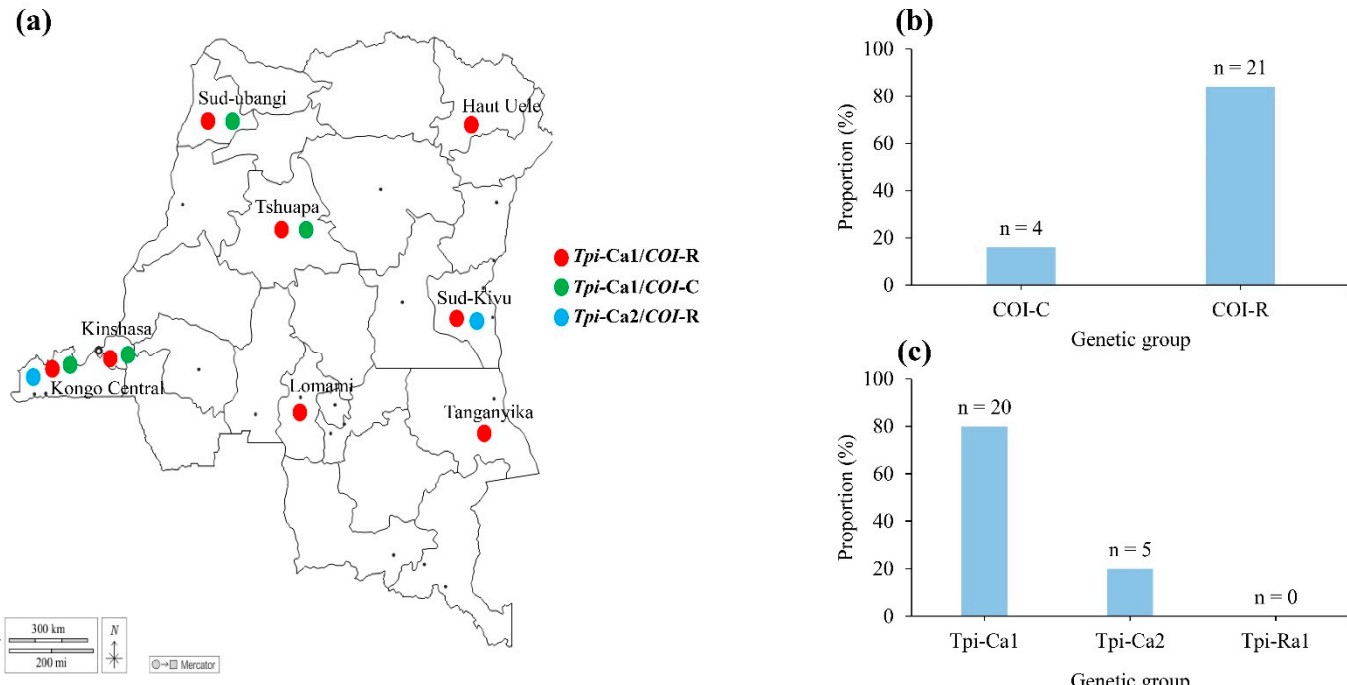

**Figure 1.** Distribution pattern of haplotypes of the fall armyworm *Spodoptera frugiperda* populations in the DRC based on collected locations (**a**), mitochondrial *COI* (**b**), and *Tpi* (**c**) partial gene markers.

### 3.2. Polymorphism Analysis

The haplotype diversity of FAW in the DRC was analyzed using 657 bp of the *COI* barcode region from individuals collected from eight provinces (Figure 1a). Our findings indicated seven polymorphic sites and a nucleotide diversity of 0.00469 (Table 2). Two distinct haplotypes (the corn and rice strains) were identified from the DRC's *COI* sequences with a haplotype diversity (Hd) of 0.324 (Table 2). Most (84%) of the *COI* sequences from this study belonged to a single rice haplotype (DRC_haplotype 1, submitted under GenBank accession number OP132901). The remaining 16% belonged to the corn strain haplotype (DRC_haplotye 2, submitted under GenBank accession number OP132898). Four sequences identified as corn strain were detected in specimens from four provinces of the DRC (Sud-ubangi, Tchuapa, Kongo central, and Kinshasa), and the rice strain sequences were detected throughout the country, suggesting that the distribution pattern of FAW haplotypes in the DRC was not region-specific.

The values of both the Fu's Fs and Tajima's D test statistic for the FAW population of the DRC were significantly positive (Table 2). Our results did not detect any nucleotide diversity within the strain populations. Genetic analysis of the FAW population in the DRC did not provide evidence of population expansion. Mismatch distribution analyses of the two strains indicated a bimodal curve, suggesting neutral evolution of FAW population in the DRC (Figure 2).

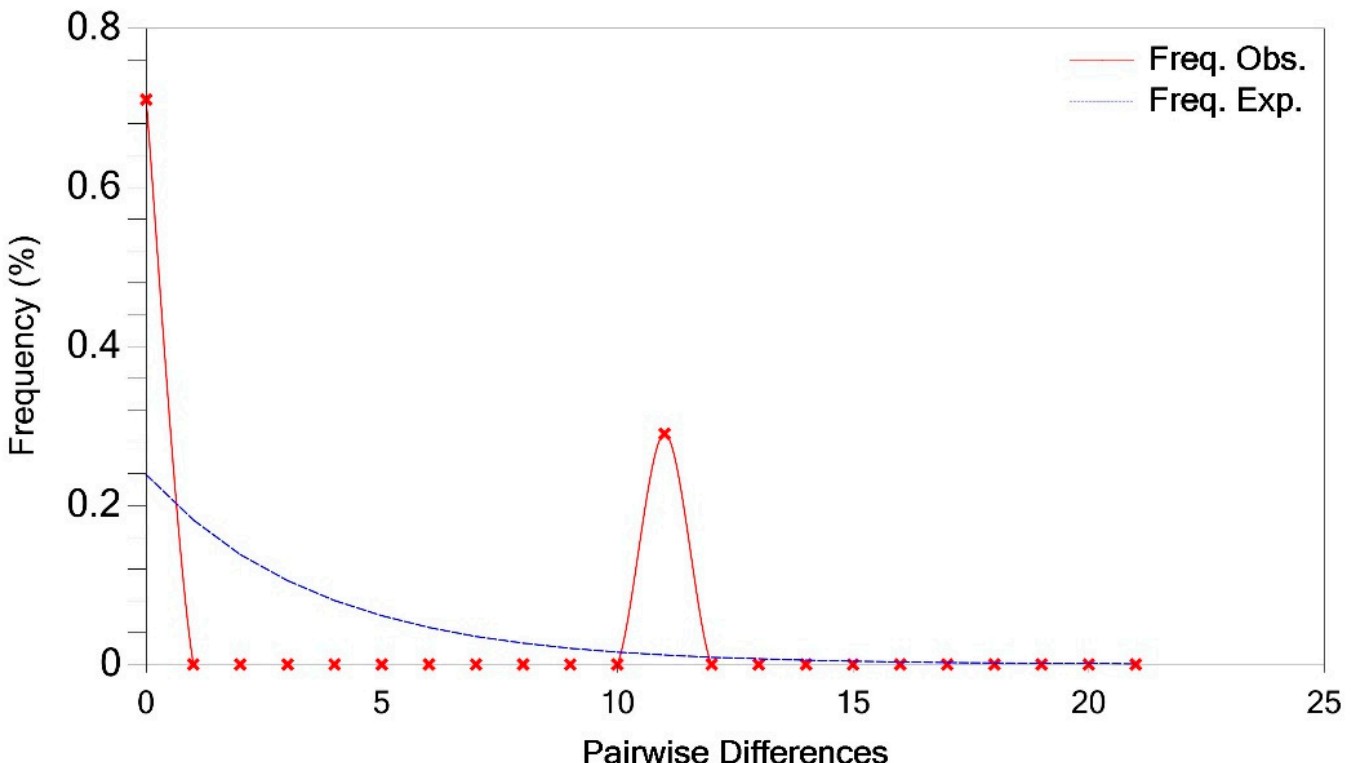

**Figure 2.** *COI* mismatch distribution curve indicating the observed (solid red line) and expected (dotted blue line) pairwise nucleotide site divergences computed with DnaSP.

**Table 2.** Summary of the genetic diversity of the fall armyworm *Spodoptera frugiperda* populations analyzed on the basis of partial *mtCOI* gene from four geographical locations.

|  | DRC | Africa | America | Asia | Total |
|---|---|---|---|---|---|
| No. of sequences | 25 | 89 | 126 | 72 | 308 |
| No. of sites | 483 | 483 | 482 | 483 | 482 |
| No. of polymorphic sites | 7 | 8 | 34 | 9 | 37 |
| No. of mutations | 7 | 8 | 38 | 9 | 41 |

**Table 2.** *Cont.*

|  | DRC | Africa | America | Asia | Total |
|---|---|---|---|---|---|
| No. of haplotypes | 2 | 3 | 29 | 4 | 32 |
| Haplotype diversity | 0.324 | 0.344 | 0.742 | 0.378 | 0.562 |
| Nucleotides diversity | 0.00469 | 0.00478 | 0.00855 | 0.00520 | 0.00735 |
| Fu's Fs statistic | 6.012 | 6.837 | −9.966 | 5.134 | −9.841 |
| Fu and Li's D × test statistic | 1.29627 | 0.47452 | −3.82406 ** | −0.08303 | −5.46527 ** |
| Fu and Li's F × test statistic | 1.14734 | 0.79287 | −3.33095 ** | 0.30287 | −4.28883 ** |
| Tajima's D | 0.53489 | 1.13421 | −1.25518 | 0.92310 | −1.28326 |

**: significant at $p < 0.02$.

### 3.3. Comparative Genetic Analyses of the FAW Population in the DRC and Three Geographic Regions

Comparative analysis of the *COI* partial gene region (483 bp) common to all the sequences, revealed haplotype numbers of 29, 3, and 4 in FAW populations from America, Africa, and Asia, respectively (Table 2). The two DRC haplotypes (rice and corn strains) were identical to the predominant rice and corn haplotypes from America (GenBank Accession: U72977.1 and U72975.1, respectively), which are most likely to be the ancestors of all haplotypes in the invaded regions. The neutrality test statistics for the DRC and African FAW populations revealed that FAW populations in these regions are still evolving neutrally relative to the American and Asian FAW populations (Table 2).

### 3.4. Comparative Phylogenetic and Haplotype Network Analysis

The phylogenetic analysis, based on the maximum likelihood method, indicated that both the rice and corn strain haplotypes from the DRC were identical to haplotypes from American, Asian and other African regions (Figure 3). Haplotype network analysis showed that there were two ancestral strain haplotypes (DRC-RS and DRC-CS) in the FAW populations of the DRC (Figure 4). The network showed the presence of the two ancestral haplotypes in the four geographical regions, with the *Tpi*-C/*COI*-R group being the dominant haplotype in the invaded regions (Africa, Asia, and the DRC). However, the distribution of novel haplotypes in America, Africa, and Asia differed significantly. Our findings suggest the 29 distinct haplotypes in America with the corn strain (*Tpi*-C/*COI*-C) as the most prevalent haplotype, whereas in the two invaded regions, the rice strain haplotype (*Tpi*-C/*COI*-R) was the most prevalent, with 3 and 4 haplotypes in Africa and Asia, respectively (Figure 3). The *COI* marker information indicated that there was no evidence of multiple introductions in the DRC.

### 3.5. Population Structure of FAW

We performed seven single AMOVA analyses, including one comparing all the geographical regions and six different combinations of groups (DRC and Africa, DRC and America, DRC and Asia, Africa and America, Africa and Asia, and America and Asia) (Table 3). The findings showed significant genetic differences among all the geographical regions (12.70%). The analysis of genetic variation among native and invasive populations indicated significant genetic differences between the native American and DRC populations (10.94%), whereas both DRC populations and those from other parts of Africa were genetically closer to the Asian populations than to American populations (Table 3).

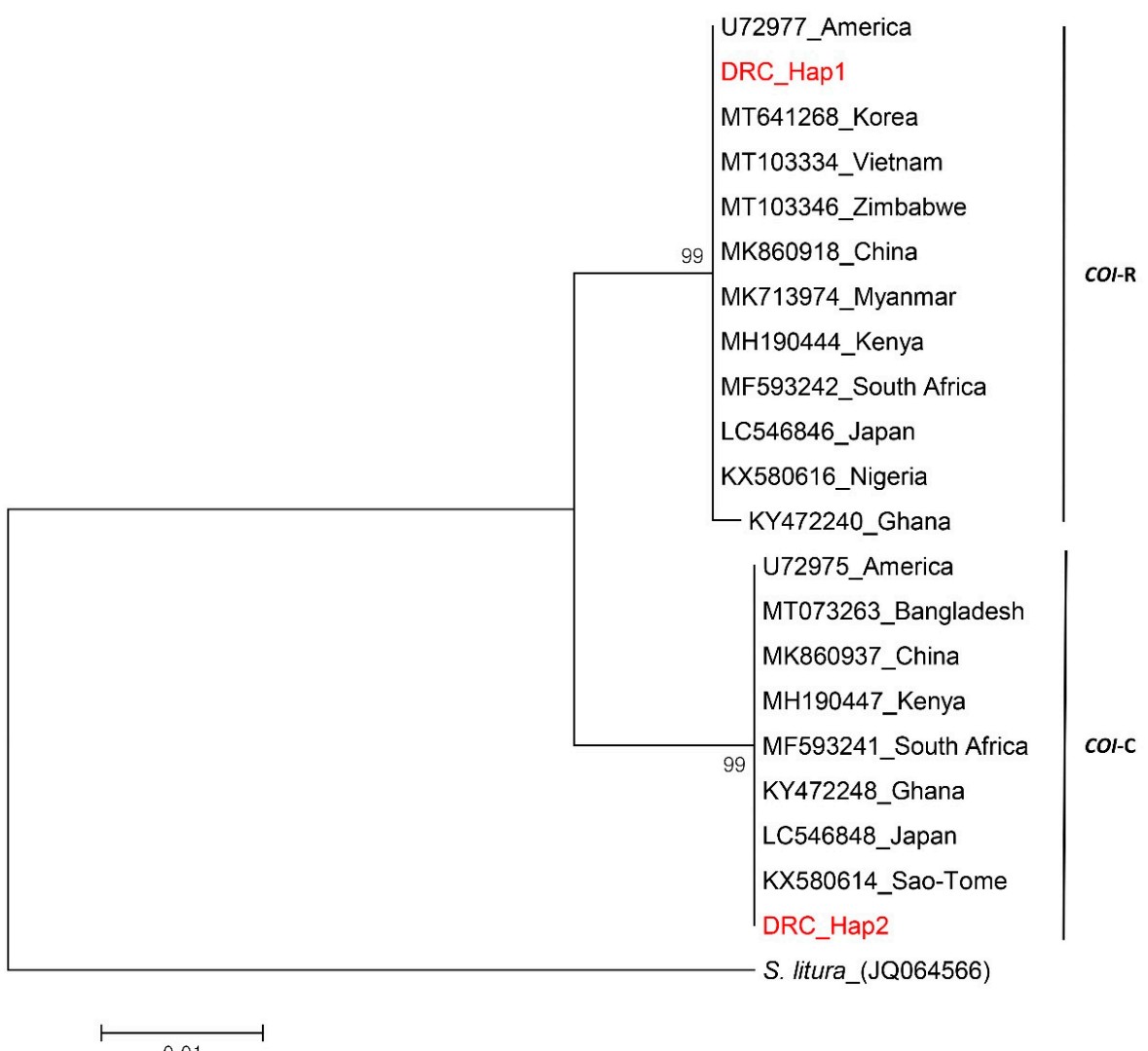

**Figure 3.** A phylogenetic tree derived from a maximum likelihood analysis comparing the two DRC *COI* haplotypes with those from *Spodoptera frugiperda* host strains of other invaded and native regions. For each phylogeny, 1000 bootstrap replicates were used to assess robustness using the Hasegawa–Kishino–Yano (HKY850) model, and gamma distribution rates of variation between sites were used to construct the phylogenetic tree in MEGA6.

**Table 3.** Results of analysis of molecular variance (AMOVA) among the four fall armyworm *Spodoptera frugiperda* geographical groups.

| Group | Source | df | SS | Variance Component | Total Variance | *p*-Value |
|---|---|---|---|---|---|---|
| All | Among groups | 3 | 71.411 | 0.2369 | 12.70 | 0.0001 |
| | Among populations within groups | 22 | 94.082 | 0.2900 | 15.54 | |
| | Within populations | 283 | 379.008 | 1.3392 | 71.76 | |
| | Total | 308 | 544.502 | 1.86629 | | |
| DRC and Africa | Among groups | 1 | 0.019 | −0.04839 | −4.33 | 0.17595 |
| | Among populations within groups | 12 | 19.429 | 0.07543 | 6.75 | |
| | Within populations | 96 | 104.744 | 1.09108 | 97.58 | |
| | Total | 109 | 124.191 | 1.11811 | | |

**Table 3.** *Cont.*

| Group | Source | df | SS | Variance Component | Total Variance | *p*-Value |
|---|---|---|---|---|---|---|
| America and DRC | Among groups | 1 | 18.325 | 0.25957 | 10.94 | 0.0001 |
| | Among populations within groups | 11 | 86.942 | 0.63554 | 26.79 | |
| | Within populations | 142 | 209.759 | 1.47718 | 62.27 | |
| | Total | 154 | 315.026 | 2.37228 | | |
| Asia and DRC | Among groups | 1 | 0.154 | −0.06807 | −4.44 | 0.1700 |
| | Among populations within groups | 10 | 22.870 | 0.11554 | 7.54 | |
| | Within populations | 81 | 120.245 | 1.48451 | 96.90 | |
| | Total | 92 | 143.269 | 1.53197 | | |
| Africa and America | Among groups | 1 | 5.473 | 0.03769 | 11.17 | 0.0001 |
| | Among populations within groups | 12 | 10.107 | 0.04366 | 12.94 | |
| | Within populations | 206 | 52.757 | 0.25610 | 75.89 | |
| | Total | 219 | 68.336 | 0.33745 | | |
| America and Asia | Among groups | 1 | 38.821 | 0.25114 | 11.51 | 0.0001 |
| | Among populations within groups | 11 | 94.217 | 0.54236 | 24.85 | |
| | Within populations | 190 | 263.859 | 1.38873 | 63.64 | |
| | Total | 202 | 396.897 | 2.18223 | | |
| Africa and Asia | Among groups | 1 | 0.132 | −0.04126 | −3.48 | 0.0400 |
| | Among populations within groups | 12 | 26.895 | 0.11284 | 9.52 | |
| | Within populations | 147 | 163.694 | 1.11356 | 93.96 | |
| | Total | 160 | 190.720 | 1.18514 | | |

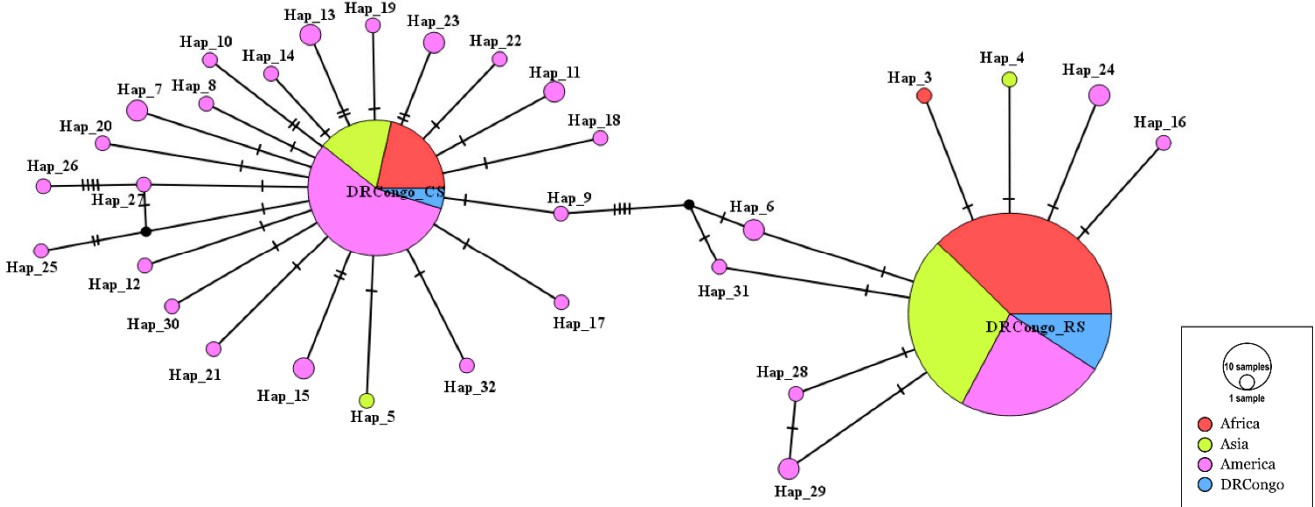

**Figure 4.** Median-joining haplotype network of the fall armyworm *Spodoptera frugiperda* mt*COI* gene partial sequences from four geographical groups (DRC, Africa, America, and Asia). Each pie represents a distinct haplotype, with the radius equal to the number of sequences belonging to that haplotype.

## 4. Discussion

This study aimed to analyze the genetic diversity and distribution of the FAW population that invaded the DRC. The findings suggest low genetic variability in the Congolese FAW population as only two haplotypes from each of the genes (*COI* and *Tpi*) were recovered. Most (84%) of the samples were *COI*-R, whereas *COI*-C occurred in 16%. These findings were consistent with those of a recent study conducted in Uganda, a neighboring country [32], and a previous report from 11 sub-Saharan African countries, including two provinces of the DRC [19]. Based on the *COI* marker, both corn and rice strains were detected in FAW specimens collected from corn fields, despite the known association of

strains to their host plant [35,36]. Similar findings have been reported in several African and Asian countries [19,22,32]. These findings suggested that the discordance between the *COI* marker and host plants may be due to FAW's plasticity in plant choice or the inability of the marker to reliably discriminate between the corn and rice strains of FAW.

The *COI*-based analysis of population genetics test statistics revealed that the FAW populations in the DRC, like those from the rest of Africa, are evolving in a neutral pattern. This neutral pattern was further supported by the absence of novel haplotypes and the low genetic diversity in FAW populations from the DRC. In contrast, the haplotype network of FAW populations in America indicates that the populations are still expanding. Thus, our findings indicate that America is still the main point of FAW population expansion. Our findings corroborate those of previous studies which also recorded that the FAW populations in Africa and America were still evolving neutrally and expanding, respectively [25,32].

Analysis of the partial sex-linked *Tpi* nuclear gene showed an 84% detection discrepancy between the *COI* and *Tpi* markers in the DRC FAW population, an observation that corroborated previous findings from other invaded regions of Africa and Asia [19,22]. In this study, we observed a dominance of the hybrid *Tpi*-C/*COI*-R individuals over the homogeneous *Tpi*-C/*COI*-C individuals among specimens collected from the corn host plant, suggesting that the *Tpi* marker is more accurate in determining the FAW–host strain association than the *COI* marker. Previously, Nagoshi et al. [17] and Nayer et al. [25] found that the hybrid *Tpi*-C/*COI*-R and the homogenous *Tpi*-C/*COI*-C were equally distributed in Central Africa, whereas in eastern and southern Africa and India, the hybrid strain predominated. Our study did not detect *Tpi*-R/*COI*-R homogenous individuals in the DRC FAW population, which occur in the western hemisphere but are rare in Africa [37]. These results are similar to those of previous studies showing that the homogeneous strain was more marginally distributed in invaded regions than the hybrid strain [19,25,32]. This hybrid strain is expected to result from the small initial interstrain mating populations explained by the admixture regularly seen during invasive events [38]. However, interstrain hybrids may have a large fitness advantage and become more prevalent in the invasive population, including in the FAW population from the DRC, eventually leading to the extinction of one or both strains in favor of more unique hybrid genotypes. These findings, combined with those of Nagoshi et al. [19] suggest that the unique African rice strain *Tpi* haplotype of the FAW found in several African regions may be rare in the DRC. In fact, Nagoshi et al. [19] detected the *Tpi*-Ra1 in the FAW specimens from the Haut-Katanga region of the DRC but not in those from the Sud-Ubangi region, which is in line with our results. These results have implications for the assessment of the crops at risk and the design of FAW management measures in the DRC. Further assessments are needed in other regions of the DRC.

Analysis of the fourth exon and intron regions of the nuclear *Tpi* gene showed the existence of two subgroups of the *Tpi*-Ca genetic group, including the predominant *Tpi*-Ca1, and minor *Tpi*-Ca2 subgroups. Further analysis showed the presence of two polymorphic variants, *Tpi*-Ca2a and *Tpi*-Ca2b, which have A or C at nucleotide 148 of Tpi-I4. The predominance of *Tpi*-Ca1a over *Tpi*-Ca2a and *Tpi*-Ca2b in invading FAW populations was also observed in Uganda [32], India [25], and several African and Asian regions [19,22].

As a highly polyphagous crop pest with a larval stage able to feed on the aerial parts of a wide range of plants, FAW can develop and establish for several generations in the DRC owing to its favorable biodiversity [39]. However, our findings combined with those of previous studies indicate that the FAW population in the DRC feeds mainly on corn plants and rarely on other plants [17,22]. This observation calls into question the nature of the hybrid genotype (*Tpi*-C/*COI*-R) detected in this study. Thus, at the molecular level, it seems that corn preference is more associated with the *Tpi* gene marker than the *COI* gene marker. This finding is not completely new because previous studies in invaded regions of Africa and Asia recorded the same genetic pattern in FAW [17,32]. These similarities in genotype features between invading populations of FAW provide evidence of a common origin

and should be used for further evolutionary studies that include the FAW whole genome sequence to better understand FAW population dynamics and subsequent dissemination in the DRC.

In summary, this study aimed to analyze the genetic diversity and distribution of the FAW population six years after the first reports of FAW invasion in the DRC. Our findings suggest that the FAW population that invaded the DRC is still evolving neutrally with a low number of haplotypes based on the *COI* gene marker. The observed low numbers of haplotypes and the hybrid nature of the FAW population in the DRC could be explained by a single introduction followed by a rapid dispersion through natural and trade-related processes. This finding combined with further studies on the migration dynamics may serve as important tools for the management of this crop pest in the DRC. Additionally, our findings showed that the nuclear *Tpi* gene marker was more accurate in determining the host association of FAW than the *COI* gene marker. Based on both the *COI* and *Tpi* markers, our study detected three genetic groups in the DRC's FAW populations, including the hybrids *Tpi*-Ca1/*COI*-R, *Tpi*-Ca2/*COI*-R, and the homogeneous *Tpi*-Ca1/*COI*-C. These results will serve as baseline resource for future studies on how the invading FAW population may change to adapt to the DRC's bio-environment and in the design of management measures. However, additional genotype surveys in other regions of the country combined with more evolutionary studies are required to refine knowledge of the FAW population dynamics and subsequent spreading routes of the pest.

**Author Contributions:** Conceptualization, M.J.M. and K.-Y.L.; methodology, M.J.M., D.M.M., G.B.B., J.C.M., H.-S.H. and K.-Y.L.; formal analysis, M.J.M.; investigation, M.J.M.; validation, K.-Y.L.; writing—original draft preparation, M.J.M.; writing—review and editing, M.J.M., D.M.M., G.B.B., J.C.M., H.-S.H. and K.-Y.L.; funding acquisition, K.-Y.L. All authors have read and agreed to the published version of the manuscript.

**Funding:** This research was supported by Basic Science Research Program through the National Research Foundation of Korea (NRF) and funded by the Ministry of Education (NRF-2016R1A6A1A05011910).

**Data Availability Statement:** Not applicable.

**Acknowledgments:** The Authors thank James Bafurume and his team for their support in sample collection. We are grateful to the farmers and extension officers for allowing different team access to their fields and guiding the team, respectively, during sample collection.

**Conflicts of Interest:** The authors declare no conflict of interest.

## Appendix A

**Table A1.** Details of FAW COI gene sequences used in the present study.

| (A) COI gene sequences from America | | | |
|---|---|---|---|
| **No.** | **GenBank Accession** | **Location** | **Year Submitted** |
| 1. | KX281221.1 | Canada | 2017 |
| 2. | U72978.1 | USA | 1996 |
| 3. | U72977.1 | USA | 1996 |
| 4. | U72976.1 | USA | 1996 |
| 5. | U72975.1 | USA | 1996 |
| 6. | U72974.1 | USA | 1996 |
| 7. | KT809294.1 | Brazil | 2018 |
| 8. | KT809293.1 | Brazil | 2018 |
| 9. | KT809292.1 | Brazil | 2018 |
| 10. | KT809291.1 | Brazil | 2018 |
| 11. | KT809290.1 | Brazil | 2018 |
| 12. | KT809289.1 | Brazil | 2018 |
| 13. | KT809288.1 | Brazil | 2018 |
| 14. | KT809287.1 | Brazil | 2018 |
| 15. | KT809286.1 | Brazil | 2018 |

**Table A1.** *Cont.*

| | | | |
|---|---|---|---|
| 16. | KT809285.1 | Brazil | 2018 |
| 17. | KT809284.1 | Brazil | 2018 |
| 18. | KT809283.1 | Brazil | 2018 |
| 19. | KT809282.1 | Brazil | 2018 |
| 20. | KT809281.1 | Brazil | 2018 |
| 21 | KT809280.1 | Brazil | 2018 |
| 22. | KT809279.1 | Brazil | 2018 |
| 23. | KT809278.1 | Brazil | 2018 |
| 24. | KT809277.1 | Brazil | 2018 |
| 25. | KT809276.1 | Brazil | 2018 |
| 26. | KT809275.1 | Brazil | 2018 |
| 27. | KT809274.1 | Brazil | 2018 |
| 28. | KT809273.1 | Brazil | 2018 |
| 29. | KT809272.1 | Brazil | 2018 |
| 30. | KT809271.1 | Brazil | 2018 |
| 31. | KT809270.1 | Brazil | 2018 |
| 32. | KT809269.1 | Brazil | 2018 |
| 33. | KT809268.1 | Brazil | 2018 |
| 34. | KT809267.1 | Brazil | 2018 |
| 35. | KT809266.1 | Brazil | 2018 |
| 36. | KT809265.1 | Brazil | 2018 |
| 37. | KT809264.1 | Brazil | 2018 |
| 38. | KT809263.1 | Brazil | 2018 |
| 39. | KT809262.1 | Brazil | 2018 |
| 40. | KT809261.1 | Brazil | 2018 |
| 41. | KT809260.1 | Brazil | 2018 |
| 42. | KT809259.1 | Brazil | 2018 |
| 43. | KT809258.1 | Brazil | 2018 |
| 44. | KT809257.1 | Brazil | 2018 |
| 45. | KT809256.1 | Brazil | 2018 |
| 46. | KT809255.1 | Brazil | 2018 |
| 47. | KT809254.1 | Brazil | 2018 |
| 48. | KT809253.1 | Brazil | 2018 |
| 49. | KT809252.1 | Brazil | 2018 |
| 50. | KT809251.1 | Brazil | 2018 |
| 51. | KT809250.1 | Brazil | 2018 |
| 52. | KT809249.1 | Brazil | 2018 |
| 53. | KT809248.1 | Brazil | 2018 |
| 54. | KT809247.1 | Brazil | 2018 |
| 55. | KT809246.1 | Brazil | 2018 |
| 56. | KT809245.1 | Brazil | 2018 |
| 57. | KT809244.1 | Brazil | 2018 |
| 58. | KT809243.1 | Brazil | 2018 |
| 59. | KT809242.1 | Brazil | 2018 |
| 60. | KT809241.1 | Brazil | 2018 |
| 61. | KT809240.1 | Brazil | 2018 |
| 62. | KT809239.1 | Brazil | 2018 |
| 63. | KT809238.1 | Brazil | 2018 |
| 64. | KT809237.1 | Brazil | 2018 |
| 65. | KT809236.1 | Brazil | 2018 |
| 66. | KT809235.1 | Brazil | 2018 |
| 67. | KJ634298.1 | Suriname | 2014 |
| 68. | KJ634297.1 | Honduras | 2014 |
| 69. | MK318422.1 | Mexico | 2019 |
| 70. | MK318420.1 | Mexico | 2019 |
| 71. | MK318377.1 | Puerto Rico | 2019 |
| 72. | MK318373.1 | Puerto Rico | 2019 |
| 73. | MK318372.1 | Mexico | 2019 |
| 74. | MK318311.1 | Mexico | 2019 |
| 75. | MK318297.1 | Dominican | 2019 |

**Table A1.** *Cont.*

| 76. | GU439151.1 | Ontario | 2018 |
|---|---|---|---|
| 77. | GU439150.1 | Puslinch | 2018 |
| 78. | GU439149.1 | Puslinch | 2018 |
| 79. | GU439148.1 | Puslinch | 2018 |
| 80. | GU439147.1 | Puslinch | 2018 |
| 81. | GU090724.1 | Puslinch | 2018 |
| 82. | GU090723.1 | Puslinch | 2018 |
| 83. | GU095403.1 | New Brunswick | 2018 |
| 84. | GU094756.1 | Puslinch | 2018 |
| 85. | GU094755.1 | Puslinch | 2018 |
| 86. | GU094754.1 | Puslinch | 2018 |
| 87. | KJ388147.1 | Quebec | 2018 |
| 88. | HM102314.1 | USA | 2016 |
| 89. | KJ641998.1 | Guano | 2015 |
| 90. | KJ641997.1 | Guano | 2015 |
| 91. | KF624877.1 | Roraima | 2014 |
| 92. | KF624876.1 | Roraima | 2014 |
| 93. | JQ559528.1 | Costa Rica | 2012 |
| 94. | JQ554012.1 | Costa Rica | 2012 |
| 95. | JQ572603.1 | Costa Rica | 2012 |
| 96. | JQ571459.1 | Costa Rica | 2012 |
| 97. | JQ547900.1 | Costa rica | 2012 |
| 98. | JQ577923.1 | Costa Rica | 2012 |
| 99. | JF854747.1 | Campina Grande | 2012 |
| 100. | JF854746.1 | Morretes | 2012 |
| 101. | JF854745.1 | Morretes | 2012 |
| 102. | JF854744.1 | Campina Grande | 2012 |
| 103. | JF854743.1 | Morretes | 2012 |
| 104. | JF854741.1 | Morretes | 2012 |
| 105. | JF854740.1 | Morretes | 2012 |
| 106. | HQ964527.1 | Massachusetts | 2012 |
| 107. | HQ964487.1 | Massachusetts | 2012 |
| 108. | HQ964486.1 | Massachusetts | 2012 |
| 109. | HQ964485.1 | Massachusetts | 2012 |
| 110. | HQ964443.1 | Massachusetts | 2012 |
| 111. | HQ964441.1 | Massachusetts | 2012 |
| 112. | HQ964442.1 | Massachusetts | 2012 |
| 113. | HQ964440.1 | Massachusetts | 2012 |
| 114. | HQ964439.1 | Massachusetts | 2012 |
| 115. | HQ964394.1 | Massachusetts | 2012 |
| 116. | HQ964393.1 | Massachusetts | 2012 |
| 117. | HQ964352.1 | Massachusetts | 2012 |
| 118. | HQ964351.1 | Massachusetts | 2012 |
| 119. | GU159435.1 | Costa Rica | 2012 |
| 120. | GU159434.1 | Costa Rica | 2012 |
| 121. | GU159433.1 | Costa Rica | 2012 |
| 122. | GU159432.1 | Costa Rica | 2012 |
| 123. | GU159431.1 | Costa Rica | 2012 |
| 124. | GU159430.1 | Costa Rica | 2012 |
| 125. | GU159429.1 | Costa Rica | 2012 |
| 126. | GU658451.1 | Alvaro Obregon | 2019 |

**(B) COI gene sequences from Africa**

| No. | GenBank Accession | Location | Year Submitted |
|---|---|---|---|
| 1. | MF593258.1 | South Africa | 2018 |
| 2. | MF593257.1 | South Africa | 2018 |
| 3. | MF593256.1 | South Africa | 2018 |
| 4. | MF593255.1 | South Africa | 2018 |
| 5. | MF593254.1 | South Africa | 2018 |
| 6. | MF593253.1 | South Africa | 2018 |

**Table A1.** *Cont.*

| | | | |
|---|---|---|---|
| 7. | MF593252.1 | South Africa | 2018 |
| 8. | MF593251.1 | South Africa | 2018 |
| 9. | MF593250.1 | South Africa | 2018 |
| 10. | MF593249.1 | South Africa | 2018 |
| 11. | MF593248.1 | South Africa | 2018 |
| 12. | MF593247.1 | South Africa | 2018 |
| 13. | MF593246.1 | South Africa | 2018 |
| 14. | MF593245.1 | South Africa | 2018 |
| 15. | MF593244.1 | South Africa | 2018 |
| 16. | MF593243.1 | South Africa | 2018 |
| 17. | MF593242.1 | South Africa | 2018 |
| 18. | MF593241.1 | South Africa | 2018 |
| 19 | MK493020.1 | South Africa | 2019 |
| 20. | MK493019.1 | South Africa | 2019 |
| 21. | MK493018.1 | South Africa | 2019 |
| 22. | MK493017.1 | South Africa | 2019 |
| 23. | MK493016.1 | South Africa | 2019 |
| 24. | MT933058 | Tanzania | 2020 |
| | MT103348 | Tanzania | |
| 25. | MT103346.1 | Zimbabwe | 2020 |
| | MT103347 | Zimbabwe | |
| 26. | KX580619.1 | Nigeria | 2016 |
| 27. | KX580618.1 | Nigeria | 2016 |
| 28. | KX580617.1 | Nigeria | 2016 |
| 29. | KX580616.1 | Nigeria | 2016 |
| 30. | KX580615.1 | Sao-Tome, | 2016 |
| 31. | KX580614.1 | Sao-Tome | 2016 |
| 32. | MT641267.1 | Uganda | 2020 |
| 33. | MF278659.1 | Tanzania | 2018 |
| 34. | MF278658.1 | Tanzania | 2018 |
| 35. | MF278657.1 | Tanzania | 2018 |
| 36. | MH190448.1 | Kenya | 2018 |
| 37. | MH190447.1 | Kenya | 2018 |
| 38. | MH190446.1 | Kenya | 2018 |
| 39. | MH190445.1 | Kenya | 2018 |
| 40. | MH190444.1 | Kenya | 2018 |
| 41. | KY472255.1 | Ghana | 2017 |
| 42. | KY472254.1 | Ghana | 2017 |
| 43. | KY472253.1 | Ghana | 2017 |
| 44. | KY472252.1 | Ghana | 2017 |
| 45. | KY472251.1 | Ghana | 2017 |
| 46. | KY472250.1 | Ghana | 2017 |
| 47. | KY472249.1 | Ghana | 2017 |
| 48. | KY472248.1 | Ghana | 2017 |
| 49. | KY472245.1 | Ghana | 2017 |
| 50. | KY472244.1 | Ghana | 2017 |
| 51. | KY472242.1 | Ghana | 2017 |
| 52. | KY472241.1 | Ghana | 2017 |
| 53. | KY472240.1 | Ghana | 2017 |
| 54. | MG993205.1 | Malawi: Sande | 2018 |
| 55. | MF197867.1 | Uganda | 2018 |
| 56. | MK493006.1 | Kenya | 2019 |
| 57. | MK493000.1 | Kenya | 2019 |
| 58. | MK492996.1 | Kenya | 2019 |
| 59. | MK493010.1 | Kenya | 2019 |
| 60. | MK493009.1 | Kenya | 2019 |
| 61. | MK493008.1 | Kenya | 2019 |
| 62. | MK493007.1 | Kenya | 2019 |
| 63. | MK493004.1 | Kenya | 2019 |
| 64. | MK493003.1 | Kenya | 2019 |

**Table A1.** *Cont.*

| | | | |
|---|---|---|---|
| 65. | MK493002.1 | Kenya | 2019 |
| 66. | MK493001.1 | Kenya | 2019 |
| 67. | MK492999.1 | Kenya | 2019 |
| 68. | MK492998.1 | Kenya | 2019 |
| 69. | MK492997.1 | Kenya | 2019 |
| 70. | MK492995.1 | Kenya | 2019 |
| 71. | MK492994.1 | Kenya | 2019 |
| 72. | MK492993.1 | Kenya | 2019 |
| 73. | MK492992.1 | Kenya | 2019 |
| 74. | MK492991.1 | Kenya | 2019 |
| 75. | MK492990.1 | Kenya | 2019 |
| 76. | MK492989.1 | Kenya | 2019 |
| 77. | MK492988.1 | Kenya | 2019 |
| 78. | MK492987.1 | Kenya | 2019 |
| 79. | MK492986.1 | Kenya | 2019 |
| 80. | MK492985.1 | Kenya | 2019 |
| 81. | MK492984.1 | Kenya | 2019 |
| 82. | MK492983.1 | Kenya | 2019 |
| 83. | MK492982.1 | Kenya | 2019 |
| 84. | MK492981.1 | Kenya | 2019 |
| 85 | MK492972.1 | Uganda | 2018 |
| 86 | MK492971.1 | Uganda | |
| 87 | MK492970.1 | Uganda | 2022 |
| 88 | MK492969.1 | Uganda | 2022 |
| 89 | MK492958.1 | Tanzania | 2020 |

**(C) COI gene sequences from Asia**

| No. | GenBank Accession | Location | Year Submitted |
|---|---|---|---|
| 1. | MT103344.1 | Bangladesh: Dhaka | 2020 |
| 2. | MT103343.1 | Bangladesh: Dhaka | 2020 |
| 3. | MT103342.1 | South Korea: Gyeongsan | 2020 |
| 4. | MT103341.1 | Viet Nam: Ninh binh | 2020 |
| 5. | MT103340.1 | Viet Nam: Ninh binh | 2020 |
| 6. | MT103339.1 | Viet Nam: Ha noi | 2020 |
| 7. | MT103338.1 | Viet Nam: Vinh phuc | 2020 |
| 8. | MT103336.1 | Viet Nam: Hanoi | 2020 |
| 9. | MT103335.1 | Viet Nam: Vinh Phuc | 2020 |
| 10. | MT103334.1 | Viet Nam: Ninh Binh | 2020 |
| 11. | MT641270.1 | South Korea: Gyeongsan | 2020 |
| 12. | MT641269.1 | South Korea: Jeju | 2020 |
| 13. | MT641268.1 | South Korea: Campus | 2020 |
| 14. | LC546868.1 | Japan: Aomori | 2020 |
| 15. | LC546867.1 | Japan: Aomori | 2020 |
| 16. | LC546866.1 | Japan: Iwate | 2020 |
| 17. | LC546865.1 | Japan: Kanagawa | 2020 |
| 18. | LC546864.1 | Japan: Chiba | 2020 |
| 19. | LC546863.1 | Japan: Fukushima | 2020 |
| 20. | LC546862.1 | Japan: Ibaraki | 2020 |
| 21 | LC546861.1 | Japan: Ibaraki | 2020 |
| 22. | LC546860.1 | Japan: Miyazaki | 2020 |
| 23. | LC546859.1 | Japan: Miyazaki | 2020 |
| 24. | LC546858.1 | Japan: Miyazaki | 2020 |
| 25. | LC546857.1 | Japan: Okinawa | 2020 |
| 26. | LC546856.1 | Japan: Okinawa | 2020 |
| 27. | LC546855.1 | Japan: Okinawa | 2020 |
| 28. | LC546854.1 | Japan: Kagoshima | 2020 |
| 29. | LC546853.1 | Japan: Kagoshima | 2020 |
| 30. | LC546852.1 | Japan: Kagoshima | 2020 |
| 31. | LC546851.1 | Japan: Kagoshima | 2020 |
| 32. | LC546850.1 | Japan: Kagoshima | 2020 |

**Table A1.** *Cont.*

| | | | |
|---|---|---|---|
| 33. | LC546849.1 | Japan: Kagoshima | 2020 |
| 34. | LC546848.1 | Japan: Kagoshima | 2020 |
| 35. | LC546847.1 | Japan: Kagoshima | 2020 |
| 36. | LC546846.1 | Japan: Kagoshima | 2020 |
| 37. | MK913648.1 | Viet Nam: Nghe An | 2019 |
| 38. | MK913647.1 | Viet Nam: Nghe An | 2019 |
| 39. | MK913646.1 | Viet Nam: Ha Noi | 2019 |
| 40. | MK860942.1 | China: Tengchong, Yunnan | 2019 |
| 41. | MK860941.1 | China: Tengchong, Yunnan | 2019 |
| 42. | MK860940.1 | China: Tengchong, Yunnan | 2019 |
| 43. | MK860939.1 | China: Tengchong, Yunnan | 2019 |
| 44. | MK860938.1 | China: Tengchong, Yunnan | 2019 |
| 45. | MK860937.1 | China: Tengchong, Yunnan | 2019 |
| 46. | MK860936.1 | China: Ruili, Yunnan | 2019 |
| 47. | MK860935.1 | China: Ruili, Yunnan | 2019 |
| 48. | MK860934.1 | China: Ruili, Yunnan | 2019 |
| 49. | MK860933.1 | China: Ruili, Yunnan | 2019 |
| 50. | MK860932.1 | China: Ruili, Yunnan | 2019 |
| 51. | MK860931.1 | China: Ruili, Yunnan | 2019 |
| 52. | MK860930.1 | China: Ruili, Yunnan | 2019 |
| 53. | MK860927.1 | China: Ruili, Yunnan | 2019 |
| 54. | MK860926.1 | China: Ruili, Yunnan | 2019 |
| 55. | MK860925.1 | China: Ruili, Yunnan | 2019 |
| 56. | MK860924.1 | China: Ruili, Yunnan | 2019 |
| 57. | MK860923.1 | China: Mangshi, Yunnan | 2019 |
| 58. | MK860922.1 | China: Mangshi, Yunnan | 2019 |
| 59. | MK860921.1 | China: Mangshi, Yunnan | 2019 |
| 60. | MK860920.1 | China: Mangshi, Yunnan | 2019 |
| 61. | MK860919.1 | China: Mangshi, Yunnan | 2019 |
| 62. | MK860918.1 | China: Mangshi, Yunnan | 2019 |
| 63. | MK713974.1 | Myanmar | 2019 |
| 64. | MN075831.1 | China | 2019 |
| 65. | MN075830.1 | China | 2019 |
| 66. | MK913645.1 | Viet Nam: Ninh Binh | 2019 |
| 67. | MT073263.1 | Bangladesh: Gazipur | 2020 |
| 68. | MT180097.1 | Pakistan | 2020 |
| 69. | OP132904.1 | South Korea | 2020 |
| 70. | MT073264.1 | Bangladesh: Bogura | 2020 |
| 71. | MT073266.1 | Bangladesh: Jamalpur | 2020 |
| 72. | MT073265.1 | Bangladesh: Rangpur | 2020 |

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
