# Peer review of "Genetic Diversity of the Fall Armyworm Spodoptera frugiperda (J.E. Smith) in the Democratic Republic of the Congo"

_agronomy, doi:10.3390/agronomy13082175_

Round 1

Reviewer 1 Report

Two fall armyworm strains, the corn strain and the rice strain, share similar morphology but exhibit a 2.09% genome divergence, making reliable strain identification crucial for field studies and monitoring fall armyworm invasions. In this study, the authors analyzed genetic variation using fall armyworm individuals collected from 8 provinces in the DRC. They compared COI polymorphisms and Tpi polymorphisms to investigate strain diversity. The results revealed that the majority of individuals showed heterogeneity, and they could be divided into two subgroups based on the fourth exon/intron sequences of the Tpi gene: TpiCa1 (80%) and TpiCa2 (20%). This finding suggests that the FAW population in the DRC exhibits some genetic diversity. Genetic variation analysis indicated significant differences (10.94%) between native American and DRC populations, with both DRC and African populations being genetically closer to Asian populations than to American populations. The research background, which motivated the study, could be better described in the abstract to provide context for the study's design and objectives. Additionally, the authors should address why data from 2019 was missing in the collection.

For better understanding, the figure legends may require additional information. In Table 2, "No" may indicate the number of polymorphic sites, and "Total" might refer to the total number of individuals analyzed.

Regarding Figure 3, it would be essential to clarify whether Bootstrap was used to test the constructed tree and, if so, how many test replicates were performed.

To explore the relationship between gene markers and plant types, additional information, such as the plant area of rice and corn in Figure 1a, could be included in the analysis.

In the Materials and Methods section, it would be helpful to include details on how the phylogenetic tree, COI mismatch distribution curve, and Median-joining haplotype network were constructed.

The source of the mtCOI data in Table 2 should be clarified, and it should be stated whether all the data were obtained through the authors' sequencing efforts.

The quality of English in this ms is fine.

Author Response

Reviewer1

Two fall armyworm strains, the corn strain and the rice strain, share similar morphology but exhibit a 2.09% genome divergence, making reliable strain identification crucial for field studies and monitoring fall armyworm invasions. In this study, the authors analyzed genetic variation using fall armyworm individuals collected from 8 provinces in the DRC. They compared COI polymorphisms and Tpi polymorphisms to investigate strain diversity. The results revealed that the majority of individuals showed heterogeneity, and they could be divided into two subgroups based on the fourth exon/intron sequences of the Tpi gene: TpiCa1 (80%) and TpiCa2 (20%). This finding suggests that the FAW population in the DRC exhibits some genetic diversity. Genetic variation analysis indicated significant differences (10.94%) between native American and DRC populations, with both DRC and African populations being genetically closer to Asian populations than to American populations.

The research background, which motivated the study, could be better described in the abstract to provide context for the study's design and objectives.

Response: The following sentence has been added in the abstract:

Lines 17-18. This study was designed to expand investigations on the genetic diversity of FAW populations in the DRC

Additionally, the authors should address why data from 2019 was missing in the collection.

Response: It was practically difficult to collect data for each year, though we planned to collect FAW data every two years. that is why the data of 2019 and 2021 are missing.

For better understanding, the figure legends may require additional information.

Response: Additional information have been provided to legends of some figures.

In Table 2, "No" may indicate the number of polymorphic sites, and "Total" might refer to the total number of individuals analyzed.

Response: Total in the table 2 indicated the overall value for the four geographic regions analyzed.

Regarding Figure 3, it would be essential to clarify whether Bootstrap was used to test the constructed tree and, if so, how many test replicates were performed.

Response: The following sentence was added to the legend of figure 3: For each phylogeny 1000 bootstrap replicates were used to assess robustness using the Hasegawa–Kishino–Yano (HKY850) model and gamma distribution rate of variation between sites were used to construct the phylogenetic tree in MEGA6.

To explore the relationship between gene markers and plant types, additional information, such as the plant area of rice and corn in Figure 1a, could be included in the analysis.

Response: The FAW specimen were all collected from corn fields which are the major crop attacked by FAW in all regions of the DR Congo.

In the Materials and Methods section, it would be helpful to include details on how the phylogenetic tree, COI mismatch distribution curve, and Median-joining haplotype network were constructed.

 Response: The following sentences were added to the section materials and methods:

Lines 131-133. Mismatch distribution curve which report the frequency of pairwise nucleotide-site differences between FAW sequences from DRC, were carried out using the constant population size model in DnaSP to further examine the demographic pattern of FAW in DRC.

Lines 145-148. Sequences were aligned and grouped within the four geographical regions using ClustalW [24] and Dnasp, respectively. The median joining network method was used to infer haplotype relationships.

The source of the mtCOI data in Table 2 should be clarified, and it should be stated whether all the data were obtained through the authors' sequencing efforts.

Response: Appendix Table A1 was added to illustrate the sequences sources.

Reviewer 2 Report

Review of Genetic Diversity of the Fall Armyworm Spodoptera frugiperda (JE Smith) in the Democratic Republic of the Congo

This paper deals with genetic diversity of populations of  the fall armyworm (FAW), Spodoptera frugiperda collected from locations in the Democratic Republic of Congo. Authors were able to establish a diversity in analysed populations and conclude that populations from Congo are similar to other African and Asian populations.

Introduction:

Authors provide the economic relevance of the fall armyworm and provide a solid background of the biology of species. Furthermore, authors indicate that different genetic markers can be used for the identification of different strains and note that populations in Africa are hybrids with relatively small genetic diversity. I would like to see, from a more theoretical perspective, how this decreased diversity can impact the crops (i.e. corn) and can this be used in some pest management strategy. Along the same line, since there is a notion of natural evolution in the Results and Discussion, I would like see the explanation this in the context of invading pest populations. What I would also like to see in this paper is the hypothesis. Currently, goals of the study are formulated ‘to provide a more representative picture of country-wide genetic structure of FAW in the DRC’ (Lines: 68-69) and ‘to compare the populations of FAW in the DRC with those from both native and other invaded regions to explore the phylogeographic pattern and relationship of FAW haplotypes in the DRC’ (Lines: 72-74). Such formulations are acceptable if there is no data on genetic structure of any population of FAW, and that is not the case here. Please make the paper more hypothesis oriented. 

Material and methods:

This section is well structured, clear and easy to follow. It demonstrates that authors are familiarised with techniques, tools and analysis that need to be used in this type of research. 

Line numbers stop after subsection 2.1.

Results:

Legend in Figure 1 does not follow the illustrated graph and map. I assume that in the legend Tpi is represented in the (c).

Table 2: it is unclear what the significance level is. In the text, authors indicate that ‘both the Fu’s Fs and Tajima’s D test statistic were significantly positive’ while in the table they indicate that ** represents significance level at p<0.02. Please clarify this. Are the numbers in the table for these parameters significant but at different levels?

Discussion:

Please elaborate how and why populations in Africa and DRC are ‘evolving neutrally’? What is the effect of such a process on populations of FAW? Furthermore, could it be that a ‘hybrid strain’ can have some competitive advantage, and if so what could it be? Connect with how low genetic variability can be used in the context of crop protection. 

Explain how the: ‘results have implications for the assessment of the crops at risk and the design of FAW management measures in the DRC.’

Generally, this paper must be improved with a hypothesis driven approach. Results on genetic diversity of  FAW can be meaningful especially for the local context.

Author Response

Reviewer2

Review of Genetic Diversity of the Fall Armyworm Spodoptera frugiperda (JE Smith) in the Democratic Republic of the Congo

This paper deals with genetic diversity of populations of the fall armyworm (FAW), Spodoptera frugiperda collected from locations in the Democratic Republic of Congo. Authors were able to establish a diversity in analyzed populations and conclude that populations from Congo are similar to other African and Asian populations.

Introduction:

Authors provide the economic relevance of the fall armyworm and provide a solid background of the biology of species. Furthermore, authors indicate that different genetic markers can be used for the identification of different strains and note that populations in Africa are hybrids with relatively small genetic diversity.

I would like to see, from a more theoretical perspective, how this decreased diversity can impact the crops (i.e. corn) and can this be used in some pest management strategy.

Response: The following sentence was added to the introduction                             

Lines 66-69. The low genetic diversity and small number of haplotypes observed in most invaded locations indicate a possible recent introduction from a common source of the FAW population and could affect the monitoring of the crop at risk in these locations.

Along the same line, since there is a notion of natural evolution in the Results and Discussion, I would like see the explanation this in the context of invading pest populations. What I would also like to see in this paper is the hypothesis. Currently, goals of the study are formulated ‘to provide a more representative picture of country-wide genetic structure of FAW in the DRC’ (Lines: 68-69) and ‘to compare the populations of FAW in the DRC with those from both native and other invaded regions to explore the phylogeographic pattern and relationship of FAW haplotypes in the DRC’ (Lines: 72-74). Such formulations are acceptable if there is no data on genetic structure of any population of FAW, and that is not the case here. Please make the paper more hypothesis oriented. 

 Response: the following sentences were added to the introduction as hypothesis of the study

Lines 74-79. The genetic characterization of FAW in the DRC during the first six years of invasion can predict changes in the populations as they rebalance and respond to pest management measures. Additionally, the comparison of the populations of FAW in the DRC with those from both native and other invaded regions can provide the phylogeographic pattern and relationship of FAW haplotypes in the DRC and would be used in understanding the possible route of the FAW population that invaded the DRC.

Material and methods:

This section is well structured, clear and easy to follow. It demonstrates that authors are familiarised with techniques, tools and analysis that need to be used in this type of research. 

Response: Thank you

Line numbers stop after subsection 2.1.

Response: corrected

Results:

Legend in Figure 1 does not follow the illustrated graph and map. I assume that in the legend Tpi is represented in the (c).

 Response: corrected

Table 2: it is unclear what the significance level is. In the text, authors indicate that ‘both the Fu’s Fs and Tajima’s D test statistic were significantly positive’ while in the table they indicate that ** represents significance level at p<0.02. Please clarify this. Are the numbers in the table for these parameters significant but at different levels?

Response: Values in the table are significant at different levels for each parameter evaluated

Discussion:

Please elaborate how and why populations in Africa and DRC are ‘evolving neutrally’? What is the effect of such a process on populations of FAW? Furthermore, could it be that a ‘hybrid strain’ can have some competitive advantage, and if so what could it be? Connect with how low genetic variability can be used in the context of crop protection. 

Explain how the: ‘results have implications for the assessment of the crops at risk and the design of FAW management measures in the DRC.’

Generally, this paper must be improved with a hypothesis driven approach. Results on genetic diversity of FAW can be meaningful especially for the local context.

Response: Some paragraph in the discussion were revised as follow

Lines 262-265: The COI-based analysis of population genetic test statistics revealed that the FAW populations in the DRC, like those from the rest of Africa, are evolving in a neutral pattern. This neutral pattern was further supported by the absence of novel haplotypes and the low genetic diversity in FAW populations from the DRC and rest of Africa.

Lines 283-287: These results are similar to those of previous studies in showing that the homogeneous strain was more marginally distributed in invaded regions than the hybrid strain. This hybrid strain was expected to result from the small initial interstrain mating populations explained by the admixture regularly seen during invasive events. Though, interstrain hybrids may have a large fitness advantage and becoming more prevalent in the invasive population, eventually leading to the extinction of one or both strains in favor of more unique hybrid genotypes.

Lines 315-322: In summary, this study aimed to analyze the genetic diversity and distribution of the FAW population six years after the first reports of FAW invasion in the DRC. Our findings suggest that the FAW population that invaded the DRC is still evolving neutrally with a low number of haplotypes based on the COI gene marker. The observed low numbers of haplotypes, and the potential interstrain hybridization could be explained by a single introduction followed by rapid dispersion through natural and trade-related processes FAW in the DRC. This finding combined with a study on the migration dynamic are important tools for the management of this crop pest in the DRC.

Round 2

Reviewer 2 Report

I am generally pleased with the changes made by the authors.